THE
EMBO
JOURNAL

# *miR-31* mutants reveal continuous glial homeostasis in the adult *Drosophila* brain

Lynette Caizhen Foo[*] iD, Shilin Song[†] & Stephen Michael Cohen[†,**] iD

## Abstract

The study of adult neural cell production has concentrated on neurogenesis. The mechanisms controlling adult gliogenesis are still poorly understood. Here, we provide evidence for a homeostatic process that maintains the population of glial cells in the *Drosophila* adult brain. Flies lacking microRNA *miR-31a* start adult life with a normal complement of glia, but transiently lose glia due to apoptosis. *miR-31a* expression identifies a subset of predominantly gliogenic adult neural progenitor cells. Failure to limit expression of the predicted E3 ubiquitin ligase, Rchy1, in these cells results in glial loss. After an initial decline in young adults, glial numbers recovered due to compensatory overproduction of new glia by adult progenitor cells, indicating an unexpected plasticity of the *Drosophila* nervous system. Experimentally induced ablation of glia was also followed by recovery of glia over time. These studies provide evidence for a homeostatic mechanism that maintains the number of glia in the adult fly brain.

**Keywords** astrocyte; glia; microRNA; neurogenesis; stem cell
**Subject Categories** Development & Differentiation; Neuroscience
The EMBO Journal (2017) 36: 1215–1226

## Introduction

Glia outnumber neurons in the human brain and are critical for providing passive trophic support for neurons. Glia also serve important active roles in sculpting the nervous system and modulating synaptic connectivity (Allen *et al*, 2012; Schafer *et al*, 2012; Chung *et al*, 2013). Glia play important roles during development and to support normal functioning of the adult brain. Despite recent advances in understanding the functions of glia, little is known about the mechanisms underlying their development or their maintenance in the adult.

Recent work has shown that *Drosophila melanogaster* glia perform functions very similar to those in mammals. Like mammalian astrocytes, *Drosophila* astrocytes encourage synapse formation (Ullian *et al*, 2001; Muthukumar *et al*, 2014; Tasdemir-Yilmaz &

Freeman, 2014). The molecular mechanisms underlying glial function in the brains of mammals are also conserved. The *Draper* signalling pathway was shown to regulate glial phagocytosis in *Drosophila* (MacDonald *et al*, 2006; Ziegenfuss *et al*, 2008), and the mammalian orthologue of *Draper, megf10*, was identified as the critical receptor mediating phagocytosis of synapses by astrocytes in the central nervous system of rodents (Chung *et al*, 2013).

In mammals, glia and neurons are produced during development from the same progenitor cells, known as radial glia (Rowitch & Kriegstein, 2010). Neural stem cells also exist in the adult brain (Doetsch *et al*, 1999), and considerable progress has been made in studying neurogenesis by adult neural stem cells. Apart from one report on the generation of adult astrocytes (Awasaki *et al*, 2008), little is known about the extent of gliogenesis in the adult brain or the mechanisms by which these progenitor cells are controlled. In this report, we provide evidence for a distinct subset of adult neural progenitor cells that are bipotent but predominantly gliogenic. These cells are identified by expression of the microRNA *miR-31a*. Mutants lacking *miR-31a* do not show a developmental defect in production of glia, but some of these cells are transiently lost in the central brain of adult *miR-31a* mutants and recover thereafter. The defect in the *miR-31a* mutant provided evidence for ongoing gliogenesis in the adult brain. Glia also recover following induced ablation in the young adult, providing evidence for a homeostatic mechanism to maintain an appropriate number of glia in the adult brain.

## Results

### Loss of astrocytes in *miR-31a* mutants

We made use of mutants from a collection of targeted miRNA knockout alleles (Chen *et al*, 2014) to examine the organization of the adult brain. Mutants lacking *miR-31a* had fewer cells expressing the glial gene *reversed polarity* (*repo*), visualized with anti-repo antibody labelling (Figs 1A and B, and EV1A and B, and Appendix Fig S1). In Canton S (CS) control animals, the number of adult glia was relatively constant with age, ranging from an average of $610 \pm 130$ at day 2 to $688 \pm 127$ at day 7 and $668 \pm 185$ at day 21 (Fig 1B and Appendix Fig S1A and B). The number of repo-expressing cells was comparable to the controls in

Institute of Molecular and Cell Biology, Singapore City, Singapore
*Corresponding author. Tel: +65 6586 9721; E-mail: lynettefoo@gmail.com
**Corresponding author. Tel: +45 3532 7326; E-mail: scohen@sund.ku.dk
†Present address: Department of Cellular and Molecular Medicine, University of Copenhagen, Copenhagen, Denmark

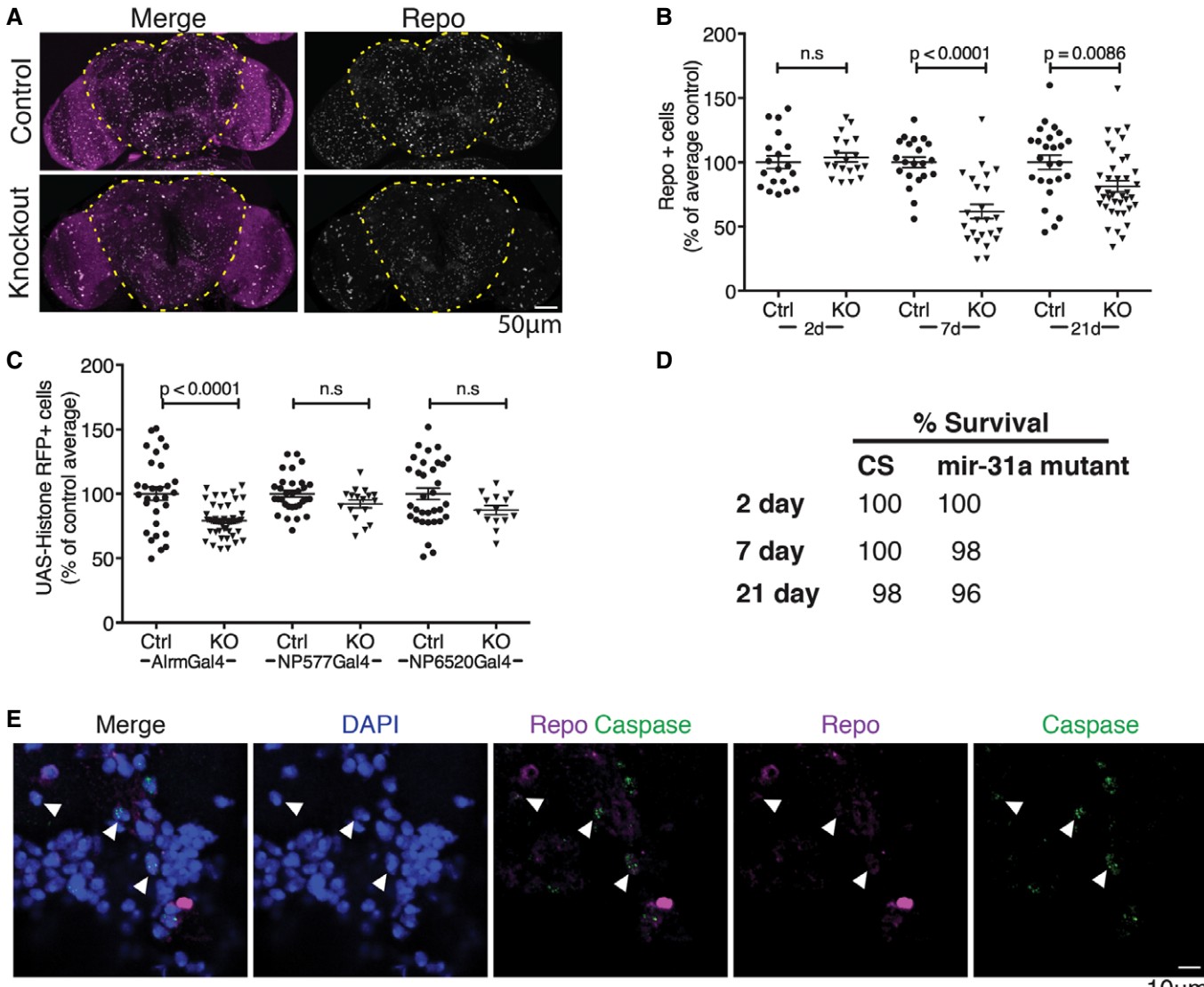

**Figure 1.  Loss of astrocytes in *miR-31a* mutant brains.**

A   Representative images of 7-days adult brains labelled with anti-repo to visualize glia and with DAPI to label nuclei (magenta). The images show maximum projections of stacks of optical sections. The central brain region in which glia were counted is outlined.

B   Number of anti-repo-positive glia in the central brain region at 2, 7 and 21 days. The number of glia is represented as a percentage of the average number of glia in central brains of controls for each age. *P*-values were determined using *t*-test (unpaired, two-tailed) comparing control and mutant at each age. Error bars represent SEM.

C   Astrocytes were labelled using *Alrm-Gal4*, and cortex glia using *NP577-Gal4* to drive *UAS-Histone-RFP*. Ensheathing glia were labelled using *NP6520-Gal4* driving *UAS-Histone-RFP*. Ctrl indicates Canton S control flies. KO indicates the *miR-31a* mutant background. *P*-values were determined using *t*-test (unpaired, two-tailed) comparing control and mutant for each driver. ns: no significant difference. Animals were analysed at 7 days post-eclosion. Error bars represent SEM.

D   Comparison of survival rate of Canton S controls and *miR-31a* mutants at 2, 7 and 21 days post-eclosion.

E   Antibody to activated caspase-3 (green) was used to visualize apoptotic cells in 4-days post-eclosion *miR-31a* mutant brains. Glia were labelled with anti-repo (purple). White arrowheads point to caspase-3-positive, repo-positive cells. Nuclei were labelled with DAPI. Images are single confocal slices.

2-day-old *miR-31a* mutants, but dropped to ~60% of the Canton S control number by day 7 (Fig 1B and Appendix Fig S1A). For ease of comparison, the data are represented as a percentage of the average of the Canton S controls. The observation that glia were present in normal numbers at day 2 suggests that the defect does not reflect a failure to produce adult glia in normal numbers during pupal development, when the majority of adult glia are born (Awasaki *et al*, 2008; Omoto *et al*, 2015). Instead, it appears that glia were lost in the mutant.

By 21 days, the number of glia in the mutant brains was on average ~80% of that in the controls, compared to ~60% at 7 days of age (Fig 1B). These observations suggested the possibility of a homeostatic mechanism by which lost glia were gradually replaced. We observed a small but significant reduction in the number of

neurons between 7-day-old Canton S controls and *miR-31a* mutants (*P* = 0.046, Fig EV1C).

To determine which type of glia were lost in the *miR-31a* mutant, we made use of Gal4 drivers to label different glial subtypes by expression of *UAS-Histone-RFP* and compared number of Gal4-positive cells in control and mutant backgrounds. *Alrm-Gal4* labels astrocytes (Doherty *et al*, 2009). *NP577-Gal4* labels cortex glia, and *NP6250-Gal4* labels ensheathing glia (Awasaki *et al*, 2008). There was no significant decrease in the number of cortex glia or ensheathing glia, but the number of *Alrm-Gal4*-expressing astrocytes was significantly reduced in 7-day-old *miR-31a* mutants (Figs 1C and EV1D, and Appendix Fig S2). Loss of *Alrm-Gal4*-expressing astrocytes accounts for approximately half of the missing repo-positive glia. The identity of the other missing glia has not been determined.

By 21 days, the number of *Alrm-Gal4* > *UAS-Histone*-RFP-expressing cells was not significantly different in mutants and control brains (Fig EV1D and Appendix Fig S2B). There was no difference in the survival of flies between Canton S controls and *miR-31a* mutant flies at 2, 7 and 21 days (Fig 1D). Thus, differences in viability cannot account for the loss and recovery of glia observed in the *miR-31a* mutants during the first 3 weeks of adult life. We detected activated caspase-3 in repo-expressing glia (Fig 1E), suggesting that glia were lost by apoptosis in the mutant and subsequently replaced.

Additional controls were performed to test whether glia might be losing *repo* expression in older animals, hence leading us to conclude incorrectly that there were fewer glia in the brain. We used flies carrying Gal80^ts and *repo-Gal4* to drive G-Trace [UAS-RFP, UAS-Flp, Ubi-p63E(FRT.Stop)GFP] in the adult. Flies were reared at 18°C until eclosion and shifted to 29°C to activate Gal4 in the newly eclosed adults. In this experiment, GFP serves as a permanent lineage tag for cells that expressed *repo-Gal4* in the adult. 94 ± 3% of GFP-expressing cells also expressed *repo-Gal4*, indicating that very few glial cells lose *repo* expression. There was also a high concordance between RFP and GFP expression indicating that few cells lose *repo-Gal4* activity. Thus, glia in the adult central brain do not lose *repo* expression at a rate that could affect the interpretation of our results (Fig EV2A).

## *miR-31a* is required in adult neural progenitors to maintain glial number

To determine where *miR-31a* activity was required, we made a microRNA sponge transgene to selectively deplete *miR-31a* in specific cell types. Depleting *miR-31a* in mature glia with *repo-Gal4* did not affect the number of surviving glia (Fig 2A and Appendix Figs S3A and S4). We confirmed that *miR-31a* activity was not detected in mature glia using a *miR-31a* sensor transgene in 2-days adult brains (Fig EV1E). Depleting *miR-31a* in post-mitotic neurons with *synaptobrevin-Gal4* also did not affect the number of surviving glia (*Syb-Gal4*, Fig 2B and Appendix Figs S3B and S4). However, depleting *miR-31a* with *elav-Gal4* (*embryonic lethal abnormal visual system*) led to a significant reduction in the number of glia (Fig 2C and Appendix Figs S3C and S4). *elav-Gal4* is expressed in post-mitotic neurons in the larval and adult stages (Osterwalder *et al*, 2001; Izergina *et al*, 2009; Li *et al*, 2013). In the embryonic CNS, *elav-Gal4* has also been shown to be

expressed in neural progenitor cells as well as in post-mitotic neurons (Berger *et al*, 2007).

The effect of depleting the miRNA with *elav-Gal4* suggested that *miR-31a* expression might be important in progenitor cells. To further explore this, we used *Inscuteable-Gal4* to label adult neuro-glial progenitors (Omoto *et al*, 2015). Depletion of *miR-31a* in *Insc-Gal4* cells led to a significant reduction in the number of glia (*P* < 0.0001, Fig 2D and Appendix Figs S3D, S4 and S5). Using the *miR-31a* sensor, we observed overlap between *Insc-Gal4* > *UAS-Histone-RFP* expression and *miR-31a*-positive (GFP-negative) in cells clustering around the antennal lobe of 2-days adult brains (Fig EV1F). We hypothesize that these overlap cells might represent an adult glial progenitor population.

Although the *miR-31a* sensor showed no overlap with mature glia expressing repo, we observed overlap with *Alrm-Gal4* > *UAS-Histone-RFP* expression in the antennal lobe (Appendix Fig S6B and C). We hypothesize that this subset of *Alrm-Gal4* cells might be immature astrocytes that have not yet begun to express repo. Alternatively, *Alrm-Gal4* might be expressed in adult glial progenitors that express *miR-31a*.

The number of glia was comparable in newly eclosed Canton S control and *miR-31a* mutant flies, and caspase labelling suggested that reduction in glial number in 7-days *miR-31a* mutant adults might be due to cell death (Fig 1). To investigate this further, we reduced the expression of *miR-31a* in progenitor cells of flies that carried one copy of the *Df(3L)H99* deficiency. This chromosomal deletion removes three apoptosis regulators, *head involution defect* (*hid*), *reaper* and *grim,* and can be used to limit apoptosis *in vivo* (White *et al*, 1994). There was no loss of glia upon expression of the miRNA sponge in this background, indicating that loss of these cells in the mutant was due to apoptosis. *Df(3L)H99* did not alter glial numbers in the control genotype where GFP was expressed instead of the *miR-31a* sponge (Fig 2D and Appendix Figs S3D, S4 and S5), indicating that the *Df(3L)H99* deficiency did not affect glial number on its own. *Df(3L)*H99 was also introduced into the *miR-31a* mutant background, where it also suppressed glial loss (Fig EV2B and Appendix Fig S7A).

The observation that glial number was normal in the mutant at 2 days but lower at 7 days suggested that the miRNA is required in the adult. To test this, we made use of temperature-sensitive *Gal80^ts* to allow temporal control of Gal4 activity. Flies carrying *Gal80^ts* together with a Gal4 driver and the *UAS-miR-31a* sponge transgene were raised at 18°C until adults eclosed and were then shifted to 29°C to allow Gal4 activity. Adult-specific depletion of *miR-31a* in *Insc-Gal4*-expressing cells led to a significant reduction in glia number compared to control animals (Fig 2E and Appendix Figs S3E and S5). Similarly, adult-specific depletion of *miR-31a* in *miR-31a-Gal4* cells led to a significant reduction in glia number compared to control animals (Fig 2F and Appendix Figs S3F and S5; the *miR-31a-Gal4* allele was generated with a vector that allows recombinase-mediated cassette exchange to replace the endogenous *miR-31a* hairpin with Gal4 sequences).

As a further test of whether the glial phenotype was due to loss of *miR-31a* expression in progenitor cells, we drove expression of *UAS-miR-31a* with *Insc-Gal4* in the *miR-31a* mutant background. Restoring *miR-31a* expression in *Insc-Gal4* cells suppressed the glial loss phenotype (Fig 2G and Appendix Figs S3G and S5). This was not observed when *UAS-GFP* was expressed instead. Expressing the

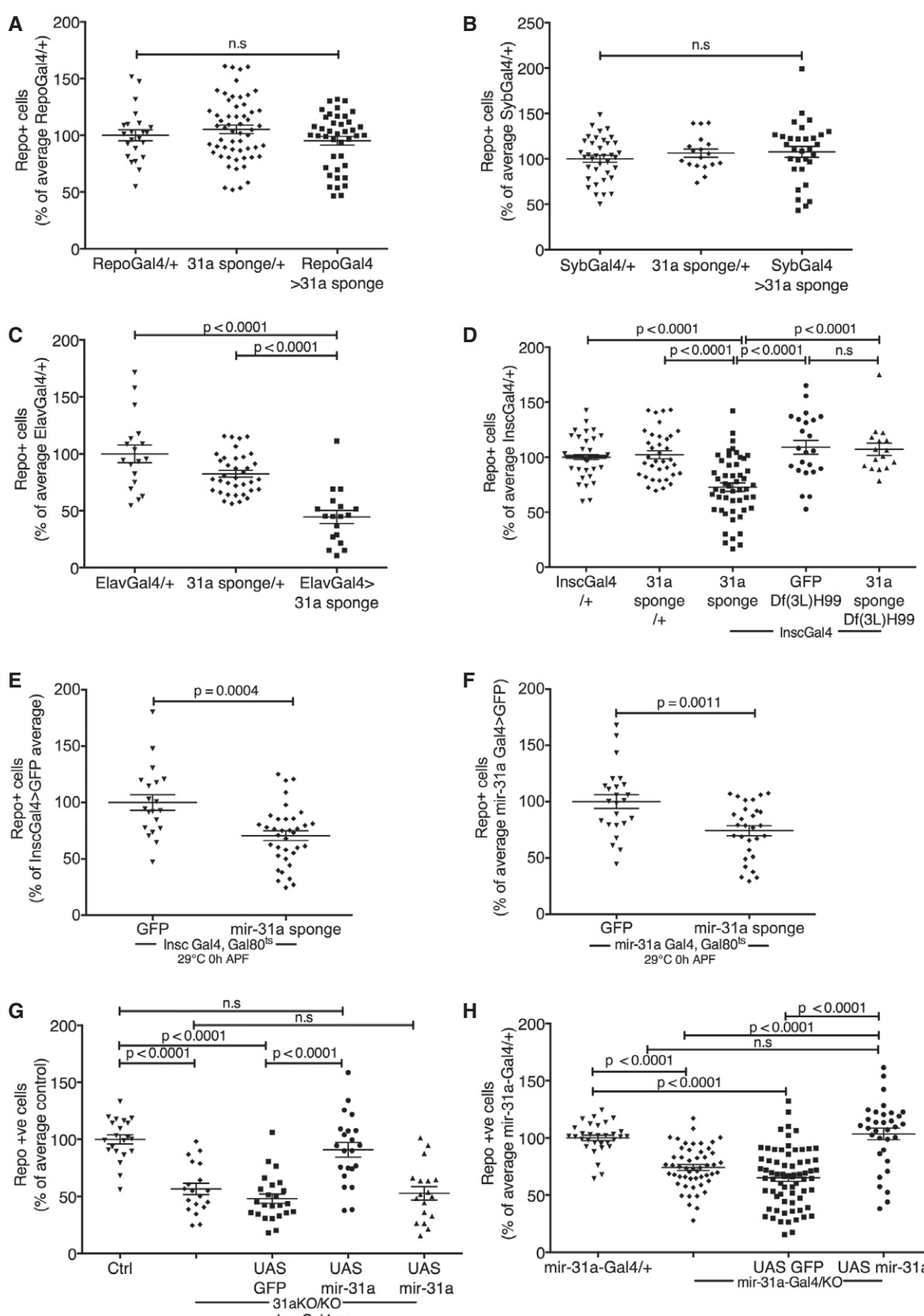

Figure 2.

**Figure 2.  *miR-31a* is required in adult glial progenitor cells to prevent glial apoptosis.**

A–D   Number of anti-repo-positive glia at 7 days of age. Glia counts in the experimental samples are represented as a percentage of the average number of glia in central brains of the Gal4/+ controls for each panel. Data were analysed using one-way ANOVA with *post hoc* Tukey analysis. Error bars represent SEM. For each Gal4 driver tested, Gal4/+ was compared with the *UAS-miR-31a* sponge transgene/+ and with Gal4 directing expression of the sponge (Gal4 > 31a sponge). (A) *repo-Gal4*, (B) *Syb-Gal4*, (C) *elav-Gal4*, (D) *Insc-Gal4, Df(3L)H99* indicate flies carrying one copy of this deletion, to limit apoptosis in the presence of the *UAS-miR-31a* sponge or *UAS-GFP*.

E, F   Number of anti-repo-positive glia following adult-specific depletion of *miR-31a*. Flies carrying *Gal80^{ts}* with *Insc-Gal4* (E) or *miR-31a-Gal4* (F) and the *UAS-miR-31a* sponge or UAS-GFP were raised at 18°C until adults had eclosed, and were then shifted to 29°C to allow Gal4 activity. *UAS-GFP* was used as a control. Flies were examined 7 days after Gal4 activation. Data were analysed using an unpaired two-tailed Student's *t*-test. Error bars represent SEM.

G, H   Number of anti-repo-positive glia in brains at 7 days. *UAS-GFP* was used as a control for expression of *UAS-miR-31a* transgene. Ctrl: Canton S control. *31a* KO/KO indicates the homozygous mutant. Data were analysed using one-way ANOVA with *post hoc* Tukey analysis. Error bars represent SEM. (G) *Insc-Gal4* was used to direct transgene expression. (H) The *miR-31a-Gal4* allele was used to direct transgene expression. Gal4/+ indicates *miR-31a-Gal4* allele *in trans* to wild-type. *miR-31a-Gal4*/KO indicates the Gal4 allele *in trans* to the deletion allele.

*UAS-miR-31a* transgene in *miR-31a-Gal4*-expressing cells also proved to be sufficient to rescue the loss of glia in the mutant (Fig 2H and Appendix Figs S3H and S6A). Furthermore, using the progenitor-specific driver *Worniu-Gal4* to direct expression of the *miR-31a* sponge recapitulated the glial loss phenotype (Appendix Figs S6D and E, and S9).

These findings suggest that the *miR-31a-Gal4* cells comprise a subset of the neuroglial progenitor population defined by *Insc-Gal4* expression. Although the miRNA does not appear to be required in mature glia, its depletion from this progenitor cell population resulted in loss of ~40% of glia in early adulthood. These observations suggest that *miR-31a* is required in the progenitor cells to support subsequent survival of newly generated glia.

### *miR-31a* targets a predicted E3 ubiquitin ligase

MicroRNAs typically downregulate their target transcripts, so it is expected that functional targets will be overexpressed in miRNA mutant tissue. Using targetscan.org, we obtained a list of predicted *miR-31a* targets and tested transcript levels with quantitative RT–PCR on RNA isolated from control and *miR-31a* mutant heads. 19 out of the 53 predicted targets tested showed an increase of more than 1.5-fold in the mutant (Table EV1). These were selected for *in vivo* functional tests using RNAi transgenes to offset the increase in target RNA levels. If overexpression of the target contributes to the mutant phenotype, then reducing its level by RNAi should decrease the severity of the phenotype.

The RNAi transgenes were expressed in the *miR-31a* mutant using *Insc-Gal4*. Only *CG16947* RNAi showed a significant increase in the number of glia in the mutant brain under these conditions (Fig 3A and Appendix Figs S7B and S8). We did not observe any reduction in the severity of the mutant phenotype with RNAi lines for any of the other *miR-31a*-predicted targets. Depletion of *CG16947* in an otherwise normal background did not affect the number of glia (Fig 3B and Appendix Figs S7C and S8). These observations suggested that the increased expression of *CG16947* in the glial progenitor population was responsible for the mutant phenotype. Consistent with this, overexpression of *CG16947* with *Insc-Gal4* in an otherwise normal background was sufficient to reduce the number of adult glia (Fig 3C and Appendix Figs S7D and S8). Comparable results were obtained when we used *miR-31a-Gal4* to direct transgene expression selectively in the smaller population of *miR-31a*-expressing cells (Fig 3D–F and Appendix Figs S7E–G, S8 and S9). Taken together, these findings indicate that overexpression of *CG16947* contributes to the *miR-31a* mutant phenotype.

The human orthologue of *CG16947* is a predicted RING finger and CHY zinc finger domain containing 1, E3 ubiquitin protein ligase (Rchy1). The *Drosophila* and human Rchy1 proteins share considerable similarity (Fig EV2C), so we used an antibody raised against a peptide including residues 87–167 of human Rchy1. Labelling with this antibody was more intense in the *miR-31a* mutant brain compared to the control, and more cells were labelled (Fig 3G and H; 17 ± 3 cells were detectably labelled in the *miR-31a-Gal4/+* heterozygous control animals compared to 88 ± 10 in the mutants, Fig EV2D). As the number of cells showing elevated Rchy1 expression exceeded the number of *miR-31a-Gal4*-expressing progenitor cells (Fig 3G), we infer that loss of the miRNA in the progenitors allows stable inheritance of Rchy1 in their progeny. Accordingly, the number of cells co-expressing Rchy1 and repo increased in the mutant (Fig 3G and H).

To ask whether expression of Rchy1 causes loss of glial cells, we drove *CG16947* expression in glia with *repo-Gal4* in an otherwise normal background. This caused a significant reduction in the number of glia in the brain compared to the *UAS-GFP* control (Fig 3I and Appendix Figs S7H and S9A). This is consistent with the hypothesis that inheritance of Rchy1 protein from the progenitor cells leads to apoptosis of their glial progeny. Additionally, we observed the presence of activated caspase in glia labelled with anti-repo in 2-days *repo-Gal4 UAS-CG16947* adult brains, indicating that overexpression causes death of glia by apoptosis (Fig EV2E).

### *miR-31a*-expressing cells are bipotent neural progenitors that are predominantly gliogenic

To explore the potential of the *miR-31a-Gal4*-expressing cells as progenitors in the adult, we performed clonal analysis using MARCM, which permanently labels their progeny with GFP. We induced clones in adult brains on the day of eclosion and labelled after 1 day with anti-elav to mark neurons (Fig 4A) or anti-repo to label glia (Figs 4B and EV3A). We observed both anti-repo-positive and anti-elav-positive GFP-expressing clones, indicating that the *miR-31a-Gal4* cells can divide in the adult to produce glial and neuronal progeny.

As another means to explore the potential of these cells, we used *miR-31a-Gal4* to drive expression of *UAS-Histone-RFP* and stained the brains with antibody to repo or elav. The perdurance of the Gal4 and RFP proteins allowed their detection together with elav protein (Fig 4C) and with repo (Fig 4D). We counted the number of *miR-31a-Gal4*-expressing cells that co-labelled with repo or elav at

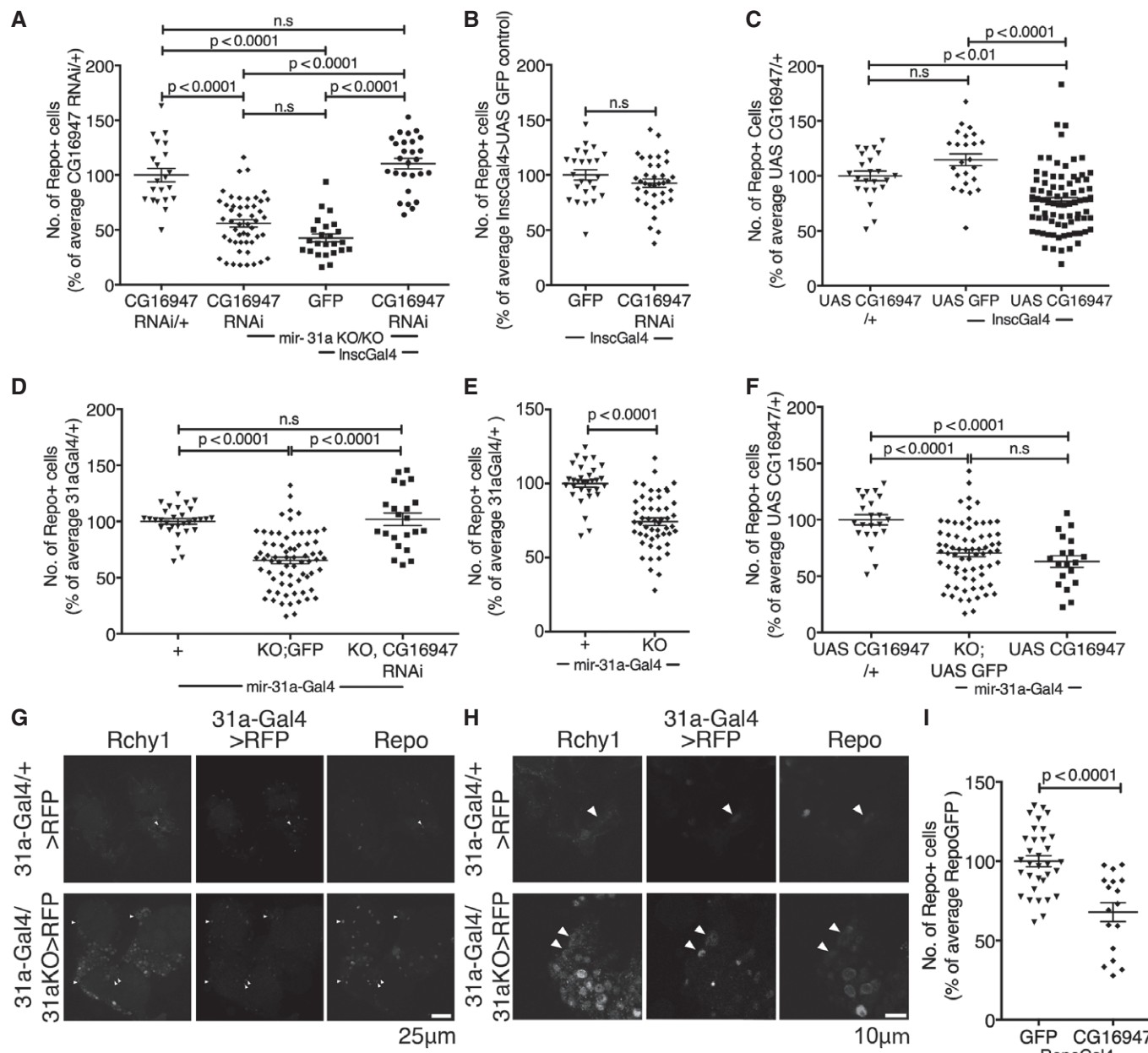

**Figure 3.  An E3 ubiquitin ligase gene, *CG16947*, is the target of *miR-31a*.**

A–F    Number of anti-repo-positive glia in adult brains at 7 days. Glia counts in the experimental samples are represented as a percentage of the average number of glia in central brains of the indicated controls. Data were analysed using one-way ANOVA with *post hoc* Tukey analysis (A, C, D, F) or with an unpaired Student's *t*-test (B, E). Error bars represent SEM. (A) The *UAS-CG16947* RNAi transgene without a Gal4 driver was used as a control. *miR-31a* KO/KO indicates the homozygous deletion mutant. A *UAS-GFP* transgene was used as a control for expression of the RNAi transgene with *Insc-Gal4* in the mutant background. ns: not significant. See also Appendix Fig S1 and Table EV1. (B) Expression of *UAS-GFP* or *UAS-CG16947* RNAi using *Insc-Gal4* cells in an otherwise normal background. (C) The *UAS-CG16947* transgene without a Gal4 driver was used as a control. *UAS-GFP* was used as a control for expression of the *UAS-CG16947* transgene with *Insc-Gal4* in an otherwise normal background. (D) All samples carried one copy of the *miR-31a-Gal4* allele. KO;GFP indicates the deletion allele and a *UAS-GFP* transgene. KO,*CG16947* RNAi indicates the deletion allele and the *UAS*-RNAi transgene to deplete *CG16947* mRNA. (E) All flies carried the *miR-31a-Gal4* allele. KO indicates the deletion allele. (F) The *UAS-CG16947* transgene without a Gal4 driver was used as a control. *UAS-GFP* in the mutant background was used for comparison to expression of *UAS-CG16947* transgene with *miR-31a-Gal4* in an otherwise normal background.

G, H    Optical sections of 7-days adult brains labelled with antibody to Rchy1 protein. *miR-31a-Gal4* was used to drive the expression of *UAS-Histone-RFP*. Glia were visualized with anti-repo. More cells expressed Rchy1, Histone-RFP and repo in the mutant brains (*miR-31a-Gal4*/KO > *RFP*) than in the heterozygous controls (*miR-31a-Gal4*/+ > *RFP*, white arrowheads). (H) Higher magnification images from the samples in (G).

I    Number of anti-repo-positive glia in 7-days post-eclosion brains from controls expressing *UAS-GFP* or *UAS-CG16947* in glia under *repo-Gal4* control. Data were analysed using an unpaired Student's *t*-test. Error bars represent SEM.

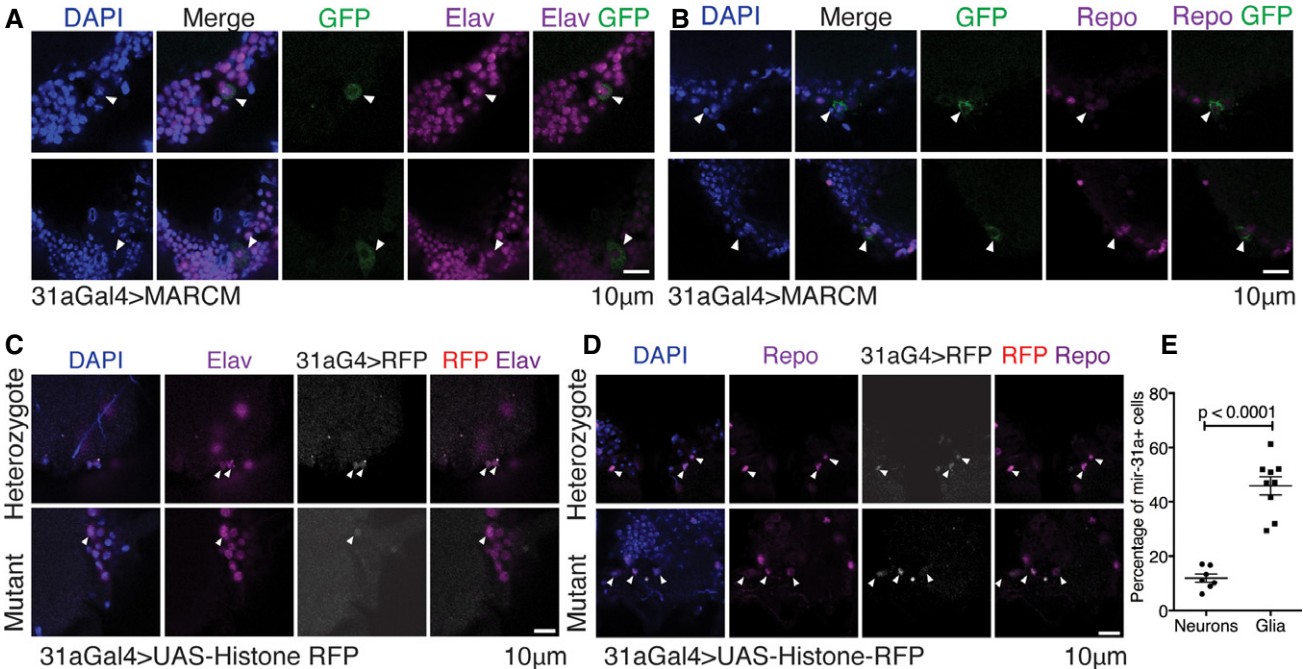

**Figure 4.** **Adult glial progenitor cells give rise to both neurons and glia.**

A   MARCM clonal analysis using *miR-31a-Gal4*. Recombination was induced by heat shock at 1 day post-eclosion to induce GFP expression in the progeny of *miR-31a-Gal4*-expressing cells. Flies were dissected the day after MARCM induction. Images show neurons expressing elav and GFP. Arrowheads indicate anti-elav-positive and GFP-expressing cells.

B   MARCM clonal analysis as in (A) using *miR-31a-Gal4*. Images show glial cells expressing repo and GFP. Arrowheads indicate anti-repo-positive and GFP-expressing cells.

C, D   *miR-31a-Gal4* (31aG4)-expressing cells were visualized with *UAS-Histone-RFP* at 1 day post-eclosion. Anti-elav was used to label neurons (C). Anti-repo was used to label glia (D). Nuclei were labelled with DAPI. Heterozygote: *miR-31a-Gal4/+*. Mutant: *miR-31a-Gal4/miR-31a*^KO allele. Figure EV3B and C shows lineage tagging using *miR-31a-Gal4, Gal80*^ts with G-Trace after 1 day or 3 days at 29°C. Arrowheads point to anti-elav-positive and UAS-Histone-RFP-positive cells (C) or anti-repo-positive and UAS-Histone-RFP-positive cells (D).

E   Percent of *miR-31a-Gal4*-expressing cells that were labelled with anti-elav (neurons) or anti-repo (glia) in 1-day post-eclosion *miR-31a-Gal4/+ > UAS-Histone-RFP* brains. Error bars represent SEM. Unpaired Student's *t*-test was used.

2 days of age. Many more of the *miR-31a-Gal4*-expressing cells were labelled with anti-repo than with anti-elav (Fig 4E), suggesting that these cells are biased towards production of glial progeny.

Next, we carried out lineage tagging using G-Trace [UAS-RFP, UAS-Flp, Ubi-p63E(FRT.Stop)GFP], which allows Gal4-regulated heritable expression of GFP (Evans *et al*, 2009). Flies carrying G-Trace with *miR-31a-Gal4* and *Gal80*^ts were raised at 18°C to keep Gal4 inactive during development. Newly eclosed adults were shifted to 29°C to allow Gal4 activity and to start lineage tagging. Brains were labelled with anti-elav and anti-repo after 1 or 3 days at 29°C. GFP-expressing progeny were found that co-expressed elav, indicating formation of neuronal progeny (Fig EV3B). GFP-expressing progeny were found that expressed repo, indicating formation of glial progeny (Fig EV3C).

Having established that the *miR-31a*-expressing cells can serve as adult progenitors for glia and neurons, we were interested in comparing their potential with those of the adult progenitors expressing *Insc-Gal4*. Using perdurance of *Insc-Gal4* and *UAS-Histone-RFP* to label recently born progeny of the *Insc-Gal4* cells, we observed neuronal progeny expressing RFP and elav, and glial progeny expressing RFP and repo. As shown in Fig EV3D, these cells represent non-overlapping populations.

Next, we combined G-Trace with *Insc-Gal4* and *Gal80*^ts and shifted flies to 29°C to activate Gal4 after eclosion. Very little background is observed in animals that were reared at 18°C until eclosion (Fig EV3E). GFP-labelled clones were detected after 1 day at 29°C. Their number had increased after 7 days, indicating ongoing proliferation of the *Insc-Gal4* progenitors in the adult (Fig EV3E). The G-Trace analysis was extended to later activation of Gal4 after 7, 14 or 21 days (Fig EV3F), after which flies were raised for an additional 7 days at 29°C before labelling brains with anti-elav or anti-repo. Approximately 50% of clones in the control flies produced elav-positive neurons at all three ages (Figs 5A and B, and EV3G). At day 7, ~20% of clones produced repo-positive glial progeny, but this dropped to ~7–8% of clones at days 14 and 21 (Figs 5C and D, and EV3H).

These data provide evidence for distinct types of multipotent progenitor cells in the adult brain. The *miR-31a-Gal4* progenitor population is predominantly gliogenic, but can also produce neurons. The *Insc-Gal4*-expressing cells are more abundant and produce more neuronal progeny than glial progeny. *Insc-Gal4*-expressing progenitor cells produced more glial progeny in the *miR-31a* mutant brains compared to the control animal brains (Fig 5D), but there was no significant change in the production of neurons (Fig 5B).

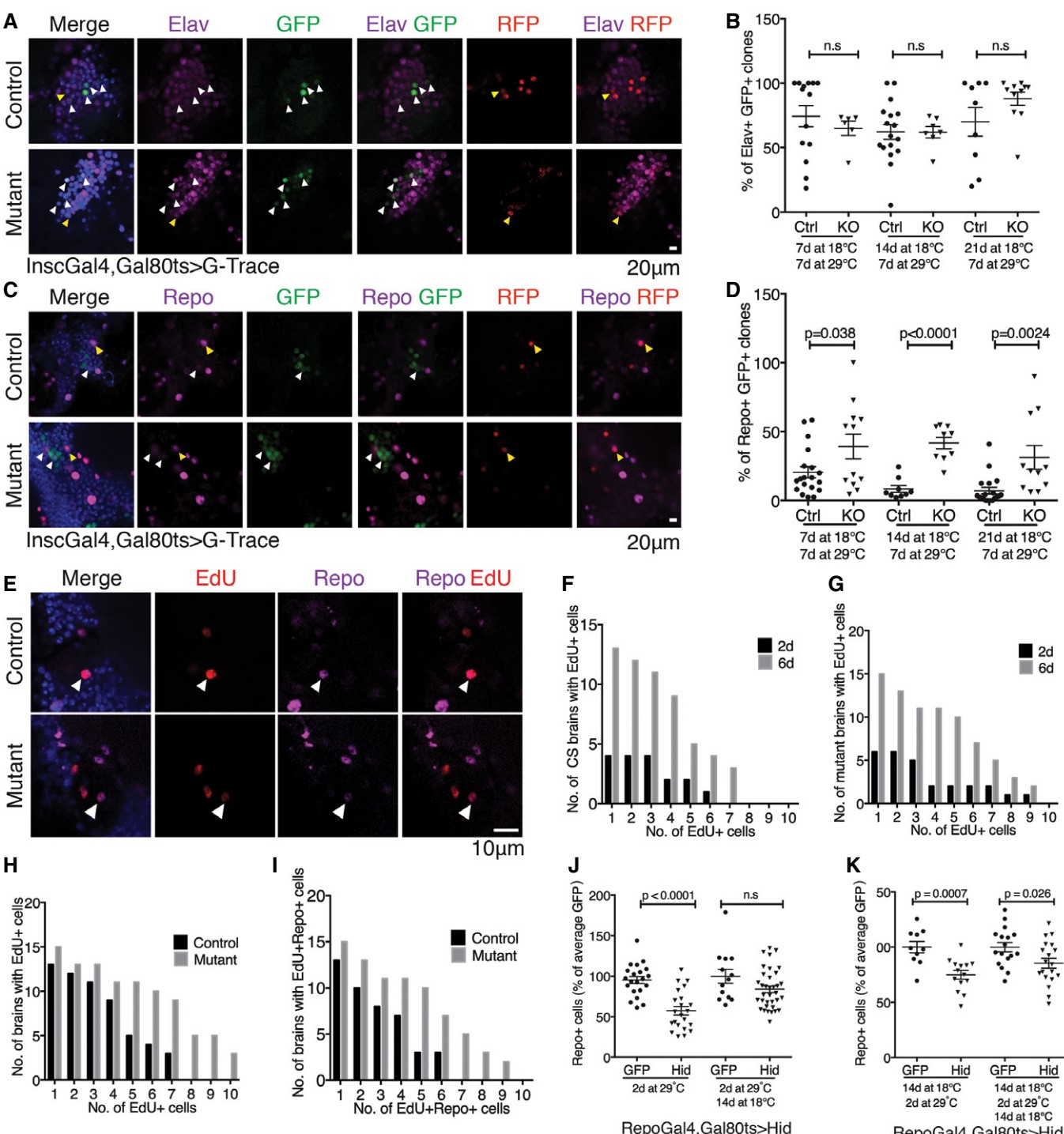

Figure 5.

## Loss of glia in the adult central brain leads to reactivation of adult progenitor cells

To further examine the proliferation of these progenitor cells, we performed a 5-ethynyl-2′-deoxyuridine (EdU) incorporation experiment to label cells undergoing DNA synthesis. Flies were fed with EdU-containing food from 24 to 54 h after eclosion. The numbers of EdU-positive cells were then counted in the central brain at the end

of the feeding period or after a 3-day chase on normal food. The number of EdU-positive cells increased between day 2 and day 6 in the Canton S control and in the mutants (Fig 5F; *P* = 0.006 for the CS controls. G: *P* = 0.001 for the *miR-31a* mutants). There were significantly more EdU-positive cells in the mutant that in the controls (*P* = 0.0002, Figs 5H and EV4A), indicating that there was more progenitor cell proliferation in mutant brains. These preparations were also labelled with anti-repo to mark glia. Figure 5I shows

**Figure 5.   Lineage tracing reveals adult glia homeostasis.**

A–D   Permanent lineage tags were induced in *Insc-Gal4*-expressing cells in the adult by controlling the expression of G-Trace [UAS-RFP, UAS-Flp, Ubi-p63E(FRT.Stop)GFP] with Gal80$^{ts}$. RFP expression reflects ongoing Gal4 activity, and GFP serves as a permanent tag for cells that have expressed Gal4. Flies were shifted to 29°C at 7, 14 or 21 days and aged for 7 days at 29°C before processing. Otherwise wild-type controls were compared with *miR-31a* mutants (KO). There was very little background leakiness of Gal4 activity in flies kept at 18°C (Fig EV3E). Data were analysed using an unpaired Student's *t*-test (two-tailed). ns: not significant. Error bars represent SEM. (A, C) Optical sections through the central brain of adult flies shifted at 14 days. (A) Neurons were labelled with anti-elav (purple). (B) Percent of clones expressing the GFP lineage tag and elav. (C) Glia were labelled with anti-repo (purple). (D) Percent of clones expressing GFP and repo.

E–I   EdU labelling was used to visualize cells undergoing DNA synthesis. Cells were counted in the central brain. (E) Images are single optical images of EdU-treated Canton S (Control) and *miR-31a* mutant brains from 6-day-old flies, 3 days after the end of EdU exposure. Note the presence of cells expressing repo and labelled with EdU (white arrowheads). (F) Number of EdU-positive cells at 2 days or 6 days after eclosion in control animals. (G) Number of EdU-positive cells at 2 days or 6 days after eclosion in mutant animals. (H) Counts of EdU-positive cells in Canton S controls and *miR-31a* mutants 6 days post-eclosion. (I) Counts of anti-repo-positive and EdU-positive in the central brains of Canton S controls and *miR-31a* mutants 6 days post-eclosion. Paired *t*-test was used.

J, K   Flies carrying *repo-Gal4* and Gal80$^{ts}$ were reared at 18°C until eclosion (J) or 14 days of age (K) to keep Gal4 inactive. This was followed by 2 days at 29°C to express *UAS-hid* or *UAS-GFP* as a control. Left: flies were examined immediately after 2 days of transgene expression. Right: flies were allowed to recover for 14 days at 18°C before processing. Anti-repo was used to visualize the glia. Data were analysed using an unpaired *t*-test (two-tailed). ns: not significant. Error bars represent SEM.

that there were more EdU-positive cells overall in the mutant brains and that a greater proportion of these cells had adopted glial identity, based on repo expression (82 ± 3% in the mutant vs 76 ± 6% in the control, *P* = 0.0004). This experiment provides additional evidence for production of new glia in the adult brain. Furthermore, it shows that the proportion of proliferating cells that make glia was higher in the *miR-31a* mutants than in the controls.

Increased proliferation and increased production of glia in the *miR-31a* mutant suggest an underlying homeostatic system that maintains glial number. If so, elimination of glia by an independent genetic means should also increase production of new glia. To test this, we used *repo-Gal4* and Gal80$^{ts}$ to transiently express the pro-apoptotic gene *hid* in adult glia. The flies were raised at 18°C until eclosion and shifted to 29°C for 2 days to express *hid* in *repo-Gal4*-expressing glia. Flies were sampled at this time or allowed to recover for 14 days at 18°C before labelling with anti-repo. After 2 days at 29°C, the number of glia was reduced by half compared to the GFP-expressing controls (Figs 5J and EV4B). There was no significant difference from the control after the 14-day recovery period, indicating recovery of the missing glia (Figs 5K and EV4B). To investigate whether glial recovery was retained in older animals, we induced hid expression for 2 days at 14 days post-eclosion. This reduced the number of glia, but the subsequent recovery was incomplete (Fig 5K and Appendix Fig S10B). These data provide evidence for a glial homeostatic mechanism that maintains the requisite number of glia in the brain. The effectiveness of glial recover after ablation was less in older flies, suggesting loss of plasticity with age.

## Discussion

### *miR-31a* acts in adult glial progenitors to support glial cell survival

Despite the critical role that glia play in the proper functioning of neurons, study of adult neural cell production has focused mainly on neurons. Much less is known about gliogenesis in the adult brain. Omoto *et al* (2015) and Awasaki *et al* (2008) recently reported that production of adult glia occurs during pupal development. In this report, we provide several lines of evidence for ongoing gliogenesis in the adult. These data include labelling the newly synthesized DNA of newly born cells in the adult showed the production of glia as well as neurons. Genetic mosaic analysis to produce clones of marked cells in the adult brain showed the

production of new neurons and new glia from progenitor cells expressing *Insc-Gal4* as well as a newly identified population of progenitors that express the *miR-31a* microRNA. Interestingly, the *miR-31a*-expressing progenitors appear to be mainly gliogenic, while those expressing *Insc-Gal4* are mainly neurogenic. Given that both progenitors can make neurons and glia, we suggest that the *miR-31a*-expressing progenitor cells may be a specialized subset of *Insc-Gal4*-expressing neuroglial progenitors. This is supported by the observation of *miR-31a* sensor activity in a subset of *Insc-Gal4* cells.

In the absence of *miR-31a* expression, glial number is affected. We noted that the number of glia is normal in young adult *miR-31a* mutants, drops sharply over several days and then recovers. This pattern implies an active process of glial turnover in the young adult brain, which is impacted by the absence of the miRNA in the progenitor population. We have provided evidence that this defect is due to the requirement for the miRNA in progenitor cells, but not in mature glia or neurons.

### Misexpression of an E3 ubiquitin ligase compromises glial homeostasis

Our findings show that *miR-31a* acts through regulation of Rchy1, a predicted E3 ubiquitin ligase. Although many transcripts are misregulated in the mutant, restoring expression of Rchy1 towards normal levels effectively suppressed the glial defect in the mutant. This implies that the overexpression of Rchy1 contributes to the failure of these cells to produce sufficient glia during the remodelling process that we observed in the young brain. In *miR-31a* mutants, Rchy1 protein was detected at elevated levels in the progeny of the *miR-31a* progenitor cells. Excess Rchy1 appears to be detrimental to the survival of the glial progeny, in that loss of glia can be suppressed by blocking apoptosis. There was only a small change in the number of neurons in adult brains of *miR-31a* mutants compared to wild-type animals. But, we note that the *miR-31a* progenitor cells are few in number compared to the predominantly neurogenic *Insc-Gal4*-expressing progenitors. Therefore, an effect of excess Rchy1 in these cells on production of neurons cannot be ruled out.

Ubiquitin-mediated protein turnover has been shown to play several important roles during CNS development in *Drosophila*. Ubiquitination and proteosomal degradation of *glial cells missing* allow embryonic glia to exit the cell cycle and begin differentiation (Ho *et al*, 2009). Additionally, neural cell fate can be controlled by ubiquitin-dependent regulation of protein translation (Hindley *et al*,

2011; Werner *et al*, 2015). Our study shows that a microRNA is responsible for regulating the level of a predicted E3 ubiquitin ligase in a neural progenitor cell, which affects the viability of its progeny. Identification of the targets of Rchy1 in neural progenitor cells may provide new insights into the control of their differentiation and survival.

### Adult glial homeostasis

Loss of glia in the *miR-31a* mutant points to a previously unidentified plasticity of the young adult brain in *Drosophila*. Our findings imply ongoing replacement of glia in the young adult brain. Furthermore, they provide evidence that the brain can detect when there are too few glia and that this can trigger progenitor cells to increase production of new glia, by increasing the proportion of their progeny that differentiate into glia. How the number of glia is monitored in the brain will be a topic for future study.

Approximately half of the glia affected in the *miR-31a* mutant were astrocytes. In this context, parallels to the mammalian brain may be interesting. In mammals, most astrocytes contact a blood vessel. Since astrocytes rely on the vasculature for survival, it is speculated that there is a matching of astrocytes to blood vessels (Foo *et al*, 2011). The fly does not have a closed circulatory system, so there are no blood vessels in the brain. There must be other mechanisms by which astrocyte number is monitored. An obvious possibility is that the need for astrocytes (or their survival) is linked to the number of neurons with which they make contact.

Two recent studies have shown that mammalian glia, specifically oligodendrocyte precursor cells (OPC) and microglia, can repopulate the brain after induced loss of the entire populations of these two cell types (Hughes *et al*, 2013; Elmore *et al*, 2014, 2015). Hughes *et al* (2013) demonstrate that OPC differentiation to oligodendrocytes occurs throughout life. As such, the homeostatic response to maintain OPCs in the brain likely reflects the need to replace the OPCs that have either differentiated or died. Their data also demonstrate that there is a continuous turnover of oligodendrocytes in the adult. Astrogenesis has been observed to occur in the prefrontal cortex of mice in response to voluntary exercise (Mandyam *et al*, 2007). Our findings provide evidence that this normal turnover of astrocytes also occurs in the *Drosophila* brain and that specific neural progenitor cells maintain an ongoing homeostatic control of astrocyte numbers in the adult *Drosophila* brain. It will be interesting to learn whether a comparable progenitor population exists in mammals to support astrocyte turnover.

# Materials and Methods

### Fly stocks

Flies were grown at 25°C or 18°C using standard fly media.

The *miR-31a*-targeted knockout mutant flies were made by homologous recombination, described in Chen *et al* (2014). A *mini-white* reporter was inserted in the place of the *miR-31a* hairpin. The RMCE mutant allele was made by targeted homologous recombination using the pRMCE vector described in Weng *et al* (2009), as

described in (Chen *et al*, 2011). The 3,923-bp 5′ homology arm was amplified with 5′gcggccgcGGCGCAGAATGAGTATTGGT3′ and 5′gcggccgcTGGTTCACAATTCACCGAAA3′. The 2,294-bp 3′ homology arm was amplified with 5′tgtgtccgtcagtacctgcaggCAGCGAGATCGAACAGAA3′ and 5′ggcgcgcctgcaggCAATCGACAGTGGTGTTTCCT3′. Following recombination, the 180-bp region encompassing the *miR-31a* hairpin was replaced with a *mini-white* cassette flanked by attP sites. The *mini-white* cassette was subsequently replaced with an attB-flanked *Gal4* cassette to generate the *miR-31a-Gal4* allele. The direction of *Gal4* insertion was verified by PCR.

To make the *miR-31a* sponge, the reverse complement of the mature *dme-miR-31a 5p* sequence was used. The sequence, TCAGCTATGACGACATCTTGCCA, was used with a point mutation made in the tenth position to change a C to an A. A fragment containing 10 copies of this sequence was synthesised with the NotI enzyme site added to the 5′ end and the XbaI enzyme site added to the 3′ end. With these two restriction enzyme sites, the *miR-31a* sponge was cloned into the pCaSpeR3 vector, downstream of EGFP. The vector was then injected into $w^{1118}$ flies and transgenic lines recovered.

To make the *miR-31a* sensor, two copies of the reverse complement of the mature sequence of *miR-31a* were cloned with XbaI and XhoI sites flanking the sequence into a plasmid downstream of GFP under the control of the tubulin promoter.

The following fly stocks were used: for astrocytes, *Alrm-Gal4* (*astrocyte leucine-rich repeat molecule*; Doherty *et al*, 2009; kind gift from M. Freeman, University of Massachusetts); for cortex glia, *NP577-Gal4;* and for ensheathing glia, *NP6520-Gal4* (Awasaki *et al*, 2008) (kind gifts from T. Lee, Janelia Farm). *Insc-Gal4* (inscuteable) (kind gift from H. Reichert, Biozentrum, University of Basel) and *UAS-hid* (kind gift from Herman Stellar, The Rockefeller University). *UAS-CD8-GFP, repo-Gal4* (glia), *elav-Gal4, CG16947* RNAi, *EGFR* RNAi, *CG1103* RNAi, *UAS-Histone-RFP, UAS-tubGal80^{ts}, FRT42D-tubulin Gal80, FRT42D* and G-Trace (Evans *et al*, 2009) fly stocks were obtained from the Bloomington *Drosophila* Stock Center. *UAS CG5021* RNAi was obtained from the Vienna *Drosophila* RNAi Centre. *UAS-CG16947* was obtained from the Harvard Exelixis Collection stocks.

### EdU labelling

Flies were reared under identical conditions at 25°C until eclosion. Male flies were picked immediately after eclosion. Flies were starved for 12 h on 1% agar, 24 h after eclosion. Thereafter, flies were fed a mixture of yeast containing 5% sucrose, 1% food colouring and 2 mg/ml of EdU (Click-iT EdU Alexa Fluor 555 Imaging Kit, Thermo Fisher, MA, USA) for 30 h. The flies were then transferred to normal fly food and dissected 72 h after the end of EdU exposure.

### MARCM

*UAS-CD8-GFP,hsFlp; FRT42D tubGal80* flies were crossed to *miR-31a-Gal4, FRT42D* flies. The flies were reared at 25°C and upon eclosion were heat-shocked at 37°C for 2 h. Thereafter, the flies were kept for an additional day at 25°C before dissection and immunostaining. A control stock of *miR-31a-Gal4* lacking FRT42D (no FRT control) was used to show that no phantom clones were observed.

## Immunostaining and analysis

Anti-repo (1:20, 8D12) and Anti-elav (1:20, 7E8A10) antibodies were obtained from the Developmental Studies Hybridoma Bank, IA, USA. Rabbit anti-RCHY1 (1:100, Millipore, HPA030339, Sigma Darmstadt, Germany), chicken anti-GFP (1:500, AB13970, Abcam, Cambridge, UK), rabbit anti-activated caspase-3 (1:500, 9578S, 9661S, Cell Signaling Technology, Danvers, MA). Alexa Fluor or DyLight highly cross-adsorbed anti-mouse and anti-rabbit secondary antibodies were used (1:1,000) (Thermo Fisher, MA, USA). Alexa Fluor anti-rat secondary antibodies were used (1:1,000). Brains were dissected in cold PBS and fixed for 20 min at room temperature with 4% paraformaldehyde and 0.2% Tween 20 in PBS. The brains were blocked in 3% bovine serum albumin in PBS for 30 min at room temperature. Brains were incubated in primary antibodies diluted in antibody buffer (150 mM NaCl, 50 mM Tris base, 1% BSA, 100 mM L-lysine, 0.04% azide, pH 7.4) over 36–48 h at 4°C and in secondary antibody for 24 h at 4°C in antibody buffer. The brains were washed in between antibodies four to five times with PBST (PBS with 0.1% Triton X) and whole-mounted in anti-fade mounting media (0.2 M Tris–HCL, pH 8.5, 2.5% *n*-propyl gallate and 90% glycerol; Stork *et al*, 2012). Images were taken using a Zeiss confocal Imager M2. ImageJ (NIH) was used with the ITCN plugin (Centre for Bio-image Informatics as UC Santa Barbara) to count the number of glial cells in the central brain. Co-localization analysis was performed with the Coloc function of Imaris (Bitplane, CT USA), and the Spots function was used to count number of cells in the brains used for co-localization analysis.

## Quantitative PCR

Total RNA was extracted with TRIzol from the frozen heads of 20–30 heads *miR-31a* KO/KO or Canton S control flies. The heads had been flash-frozen in liquid nitrogen and were not thawed prior to extraction. For target RNA screening, the RNA was treated with DNase using the RNA cleanup protocol from the RNeasy mini kit (Qiagen). RNA was quantitated via spectrophotometry (NanoDrop, Thermo Scientific). The 260/280 ratio of the RNA samples was ~2.0. Samples ranged in concentration from 191 ng/µl to 424 ng/µl in 35 µl of water. 1 µg of total RNA for each sample was converted to cDNA. First strand DNA synthesis was done with Superscript RT-III (Invitrogen) and 1 µl of a 50 mM stock of oligo dT primers. The oligo dT primers were first incubated with 1 mM of dNTP mix and 1 µg of RNA diluted in water at 65°C for 5 min in a reaction volume totalling 13 µl. The samples were then incubated on ice for 1 min after which a reaction mixture of Superscript RT-III, DTT and RNAase OUT was added to each sample. The samples were then incubated at 50°C for 50 min and the reaction terminated at 85°C for 5 min. All steps were performed in a PCR machine.

Quantitative RT–PCR was done with Fast SYBR Green Master Mix (#4385612, Applied Biosystems) in the Applied Biosystems 7500 Fast Real-Time PCR machine. The data were normalized to the reference genes, actin (F 5′ GCTTCGCTGTCTACTTTCCA, R 5′ CAG CCCGACTACTGCTTAGA), GAPDH (F 5′ CCACTGCCGAGGAGGTC AACTAC, R 5′ ATGCTCAGGGTGATTGCGTATGC) and RP49 (F 5′ G CTAAGCTGTCGCACAAA, R 5′ TCCGGTGGGCAGCATGTG). PCR primer efficiency was determined for each primer pair, and primers were only used if the efficiency exceeded 85%. Data represent the average of three independent biological replicates.

| Quantitative RT–PCR conditions | |
|---|---|
| Holding stage | Step 1: 95°C 20 s |
| Cycling stage | Step 2: 95°C 3 s |
| | Step 3: 60°C 30 s |
| Return to step 2: 40 times | |
| Melt curve stage | Step 4: 95°C 15 s |
| | Step 5: 60°C 60 s |
| | Step 6: 95°C 15 s |
| | Step 7: 60°C 15 s |

## Box-shade analysis

http://www.ch.embnet.org/software/BOX_form.html was used to compare the aligned protein sequences of *Drosophila melanogaster* CG16947 and human RCHY1.

## Statistics

Prism 6 (GraphPad, CA, USA) was used to conduct statistical analysis. Unpaired student's *t*-test and one-way ANOVA with *post hoc* Tukey analysis were used. For *t*-tests, Cohen's *d* (1.6) was used to calculate the effect size; sample size for experiments presented exceeded this value. Refer to scatter plot graphs for number of *n* per experiment. The *F*-test was used to compare the variance between two samples. For one-way ANOVA tests, Bartlett's test was used. For EdU studies, paired test was used. Cell counts for Figs 1–3 were done with the experimenter blinded. All brains dissected from live animals on the day of collection were included in the analysis; no brains were excluded from analysis. For the EdU experiments, only flies that had an ingested comparable amount of EdU as indicated by the redness of the belly from the red food dye added to the yeast paste were dissected and analysed. SEM values and sample size for all experiments are in Table EV2.

**Expanded View** for this article is available online.

## Acknowledgements
This work was supported by core funding from IMCB. SC is supported by a grant from the Novo Nordisk Foundation (NNF12OC0000552). We thank Niva-sini Kumar and Tan Kah Junn for technical support.

## Author contributions
LCF performed the experiments. LCF and SMC designed the experiments and wrote the manuscript. SS contributed to producing the *miR-31a* RMCE allele.

## Conflict of interest
The authors declare that they have no conflict of interest.

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
