## [Review Process File · The EMBO Journal]

Manuscript EMBO-2016-95861

miR-31 mutants reveal continuous glial homeostasis in the adult *Drosophila* brain

Lynette Caizhen Foo, Shilin Song, Stephen Michael Cohen

Corresponding author: Stephen Cohen, University of Copenhagen & Lynette Foo, Institute of Molecular & Cell Biology, Singapore

Review timeline:

Submission date:	11 October 2016
Editorial Decision:	16 November 2016
Revision received:	11 January 2017
Editorial Decision:	07 February 2017
Revision received:	09 February 2017
Accepted:	13 February 2017

Editor: Karin Dumstrei

Transaction Report:

1st Editorial Decision

16 November 2016

Thanks for submitting your manuscript to The EMBO Journal. Your study has now been seen by three good experts and their comments are provided below.

As you can see from the reports, the referees find the analysis interesting and suitable for publication here. However, they also raise a number of constructive comments that should be addressed for publication here. No further reaching experiments are needed it is matter strengthening the key findings/conclusions. The raised issues are clearly outlined below and I will not repeat them here. Let me know if we need to discuss any specifics further.

I should add that it is EMBO Journal policy to allow a single major round of revision only and that it is therefore important to address the key issues at this stage. You can use the link below to upload the revised version.

When preparing your letter of response to the referees' comments, please bear in mind that this will form part of the Review Process File, and will therefore be available online to the community. For more details on our Transparent Editorial Process, please visit our website: http://emboj.embojpress.org/about#Transparent_Process

We generally allow three months as standard revision time. As a matter of policy, competing manuscripts published during this period will not negatively impact on our assessment of the conceptual advance presented by your study. However, we request that you contact the editor as soon as possible upon publication of any related work, to discuss how to proceed. Should you

foresee a problem in meeting this three-month deadline, please let us know in advance and we may be able to grant an extension.

I look forward to seeing the revised version.

REFEREE REPORTS

Referee #1:

The manuscript presents an unexpected role of mir-31a in regulating astrocyte cell number in the adult *Drosophila* brain. In mir-31a mutants astrocytes undergo apoptosis but are replaced after 3 weeks of life. Based on cell type specific depletion experiments, mir-31a appears to be required in progenitor cells and is dispensable in differentiated glial cells. Gal80ts experiments demonstrate that mir-31a is required in adult progenitor cells to suppress cell death of the differentiated glia suggesting that mir-31a normally prevents an accumulation of a toxic factor that leads to apoptosis of the differentiated cells.

In a next step the authors identified a mir-31a target gene (CG16947) that when overexpressed in the the progenitor cells causes a reduction of glial cell number. The authors generated an antibody against the CG16947 protein and report increased expression in a mir-31a knockdown background. The normal function of CG16947 is, however, not revealed.

This paper reports two in principle interesting findings. First the authors demonstrate that mir-31a is required in glial progenitor cells to suppress apoptosis of the differentiated astrocytes and second the authors report an apparent reactivation of progenitor proliferation to „regenerate" the missing astrocytes.

The mir-31a mutant phenotype results in a transient loss of 40% of all glial cells. This means that more than 200 glial cells die, an amazing cell number which is not really addressed. I cannot imagine that changes in astrocyte numbers can explain this reduction and unfortunately, Figure 1 gives only relative numbers. The notion that mir-31a is required in the astrocyte progenitors to allow survival of the differentiated cells is not really solved. In fact, the adult depletion of mir-31a in adult progenitor cells also results in the loss of adult glial cells - but at this developmental stage all glial cells have been born already. How can the cell death be explained? The expression of mir-31a is not well documented and the images are not instructive. It would be indeed interesting to see the localization of astrocyte progenitors are located. The generation of adult astrocytes was previously analyzed by Awasaki et al., but this paper is unfortunately not discussed. The present study only uses Repo staining to determine cell numbers and does not address the shape of the astrocytes. In a previous report, Stork et al., (2014) had demonstrated that upon induction of cell death in astrocytes, the surviving cells expand their size. Is this observed in mir-31a mutants as well? And is this process reversed in the regeneration phase?

Referee #2:

In "Defects in astrocyte production in mir-31a mutants unveil glial homeostasis in the adult *Drosophila* brain," Foo and colleagues describe a mechanism for continued glial turnover in the mature fly brain by newly identified mir-31a-positive neuroglial progenitors. The authors demonstrate that reduced mir-31a function in global mutants and by RNAi KD in progenitor cells resulted in transient loss of brain glia. While mir-31a was not required for initial production of glial cells, they found that glial cells were lost over time due to apoptosis, and that cell death occurred due to impaired post-transcriptional regulation of the E3 ubiquitin ligase Rchyl1 by mir-31a. Interestingly, glial cell loss was followed by a period of biased glial cell production from mir-31a-positive cells, suggesting a homeostatic mechanism by which glial cell numbers are closely monitored and maintained within the adult brain.

Overall, the data that the authors present is clear, and the importance of the work is quite high. The authors should be commended for the depth of their analysis. Yet there are several important changes/additions necessary before it is suitable for publication in EMBO J.

Major comments:

1. What is the "cell of origin" for the replacement glia? - glia? neurons? progenitors? other? The evidence that they come from mir-31a-gal4 positive cells is not sufficient to identify the cell type that produces these late-born glia. For example, on page 16 it says "production of new neurons and new glia from progenitor cells" - what is the evidence that these new cell come from neuroblasts (or any known progenitor)?

2. In general, the authors provide many graphs to support their conclusions, but not enough representative images of these data. To feel confident in their conclusions, representative images for the following experiments should be displayed. If space limitations are a problem, some of the images could be included as supplemental figures.

- a) Figure 1C. The paper highlights that these data suggest a mechanism for adult astrocyte homeostasis that hasn't been clearly demonstrated before, so it is essential to show that astrocytes are specifically affected, using an astrocyte specific marker.
- b) For images comparing the mutant phenotype to loss of glia due to use of mir031a sponge in *insc-Gal4* (Fig. 2D) and *mir-31a-Gal4* (Fig. 2F) animals, as they are used as proxies for the global mutant. Because the data is always presented as % of the control, it is not possible to determine how well the cell-specific KDs reflect the mutant phenotype.
- c) For experiments where the authors suppress the mutant phenotypes either by blocking apoptosis (Fig. 2D) or by providing exogenous mir-31a (Fig. 2H)
- d) Fig. 3A, C to definitively show mutant phenotype in response to CG16947 manipulations

3. The authors should use a "progenitor specific" *gal4* line, such as *wor-gal4* in addition to *insc-gal4* (which is expressed in some glia; Omoto et al., 2015). Showing both give the same phenotype, for at least a few experiments, would help prove that the effects are neuroblast specific.

4. page 16, are the EdU+ glia endoreplicating or forming de novo?

Minor issues

- please provide more information in Table S1 (e.g. p-values, primers used for all genes) and in the methods regarding their qPCR conditions, standardizations and results, which should follow MIQE guidelines (PMID 19246619)
- authors should not use the term 'astrocytes' unless cell identity is confirmed with an astrocyte marker (e.g. *alrm-gal4*)
- The authors show that overexpression of CG16947 results in reduced glial cell number, but they don't directly show that this is due to apoptosis (as was shown for loss of mir-31a). Please stain for a cell death marker to show that overexpression of *Rachyl* mimics the mir-31a mutant phenotype.
- Whenever presenting data, please include +/- SD or SEM. For example, on page 5, "dropped to approximately 62% of Canton S controls" how many n's and what is the range of glial numbers are not shown.
- Throughout the text and figure legends, please be more clear about whether cells were counted by reporter expression or by antibody stain. For example, the authors use both *repo-Gal4* and anti-Repo, and it's not always clear what method was used for counting
- Fig. 4E, please state what animals and what time point was used for this quantification
- Please label each figure panel with the time point analyzed
- The wording in the discussion is at times too strong based on the data the authors provide. For example, on page 17, the authors intimate that the mir-31a progenitors are specifically astrogenic. Although the data that they provide shows that glial survival is specifically affected by loss of mir-31a, these data do not preclude that mir-31a progenitors also give rise to cortex and ensheathing glia and that they simply do not have a survival phenotype. 2) For example, on page 19, glial turnover has been shown in the healthy brain during learning/memory tasks in rodents (reviewed in PMID 4634839).
- the authors should carefully review and edit genetic nomenclature throughout the text and figures (e.g. use of lowercase italics whenever genes and transgenes are used for *Drosophila* versus rodent genes)

Very minor issues:

- Consistency with labeling in the figures
- Make sure to define transgenes when they're first presented. For example, *repo-Gal4* is first

defined on page 7, but introduced on page 6

- Typo on page 11 "labeled after 1 day with anti-Repo to mark neurons (Figure 4A) or glia (Figure 4B, S2A). The authors used Elav to mark neurons, Repo to mark glia
- Fig. 5J+K are cutting off parts of the labels for Fig. 5H + I

Referee #3:

General Summary:

This paper by Foo and colleagues describes a novel progenitor pool in the young adult *Drosophila* brain that is identifiable by microRNA mir-31a expression. These progenitors give rise largely to glial cells, likely astrocytes. In the absence of mir-31a, additional adult progenitors restore glial cells to normal numbers over the course of weeks. This provides the first description of a homeostatic mechanism in the adult *Drosophila* brain to maintain sufficient number of glia. These results are novel and very exciting and will provide important new insight into microRNA and ubiquitin ligase-mediated regulation of glial cell numbers in adults, which will be of broad interest to many in the field of glial cell biology. Overall, the manuscript is well written and the conclusions largely supported. Some thoughts and comments to improve the paper are provided below:

Major points:

- 1 . Figure 1A, the loss of Repo-positive cells is apparent. However, because astrocyte cell bodies are specifically located at the edge of neuropil regions, it's a little surprising that the Repo reduction appears consistent (broad) throughout the entire brain. Perhaps due to the particular image that was selected? It would be nice if confocal images were provided for some additional key experiments throughout the paper. For example, images corresponding to the Fig 2E-H. This would provide another opportunity to potentially observe the spatial distribution of changes in Repo expression throughout the central brain.
- 2 . As written, the manuscript seems to make an abrupt transition from using *alm-Gal4* to *Insc-Gal4* to visualize/manipulate glia. Is there experimental evidence showing that *alm+* positive cells do not give rise the same progenitors that *Insc+* (proliferative) cells do? *Insc* may not be specific to astrocytes. It would nice if a subset of key expts were repeated co-labeling with the GABA transporter *Gat* - or the excitatory amino acid transporter *EAAT1* (aka *GLAST*) - to show that the newly produced cells that contribute to recovery of the glial population in *mir31a* mutants are indeed astrocytes.
- 3 . Repo-driven expression of the *mir31a* sponge did not alter the number of glial cells. Repo appears to co-express with *mir31a* drivers quite early (during the first week). How do the authors reconcile the fact that there was no phenotype with *RepoGal4* -> *mir31a* sponge expression?
- 4 . Using heterozygote H99 mutant animals may influence apoptosis during development. In Figure 2D, the authors show that the reduction of Repo-positive cells in *31a* sponge flies is rescued in H99 heterozygotes. Might the H99 hets already have higher levels of glia (and perhaps other cells?) Including an H99 heterozygotes is an essential control in this experiment. Also, confirming that apoptosis is altered with an activated caspase-3 stain would be nice addition to round out this part of the model.
- 5 . The model of astrocyte homeostasis is intriguing one, and this paper presents the first description of this mechanism in adult *Drosophila*. The authors state that "turnover of astrocytes has not yet been observed in healthy brains of adult mammals", but I don't believe this is true. For example, Sohn et al., (*J Neuroscience* 35(9): 3756-63, 2015) describe astrocyte production balanced by apoptosis in adult mice.

Minor points:

- 1 . State N values for experiments.
- 2 . Missing word: Depleting *mir-31a* in mature glia with *Repo-Gal4* (reversed polarity, Figure 2A)

or in post-mitotic neurons WITH synaptobrevin-Gal4 (Syb-Gal4, Figure 2B) did...

3 . Change sentence: "Adult flies were heat-shocked the day of eclosion to induce clones and the brains were labeled after 1 day with anti-Elav or anti-Repo to mark neurons (Figure 4A) or glia (Figure 4B, S2A)."

4 . Fig 4C and D are incorrectly labeled "31aGal4,G80ts>G-Trace". There is no G-Trace in this image I think.

5 . Fig 5E - are mutant and control different magnifications?

6 . Fig 5H - What time point was this and where were the EdU cells counted? The whole brain?

7 . I'm assuming the experiment shown in Supplemental Figure 1E was done in a mir-31a mutant background? It doesn't explicitly say that in the Results or Figure/Figure legend.

8 . Most of the images in Fig 3G,H are very difficult to see. All channels can be converted and grayscale since there are no merged images.

1st Revision - authors' response

11 January 2017

Referee #1:

The manuscript presents an unexpected role of mir-31a in regulating astrocyte cell number in the adult Drosophila brain. In mir-31a mutants astrocytes undergo apoptosis but are replaced after 3 weeks of life. Based on cell type specific depletion experiments, mir-31a appears to be required in progenitor cells and is dispensable in differentiated glial cells. Gal80ts experiments demonstrate that mir-31a is required in adult progenitor cells to suppress cell death of the differentiated glia suggesting that mir-31a normally prevents an accumulation of a toxic factor that leads to apoptosis of the differentiated cells.

In a next step the authors identified a mir-31a target gene (CG16947) that when overexpressed in the the progenitor cells causes a reduction of glial cell number. The authors generated an antibody against the CG16947 protein and report increased expression in a mir-31a knockdown background. The normal function of CG16947 is, however, not revealed.

This paper reports two in principle interesting findings. First the authors demonstrate that mir-31a is required in glial progenitor cells to suppress apoptosis of the differentiated astrocytes and second the authors report an apparent reactivation of progenitor proliferation to „regenerate" the missing astrocytes.

The mir-31a mutant phenotype results in a transient loss of 40% of all glial cells. This means that more than 200 glial cells die, an amazing cell number which is not really addressed. I cannot imaging that changes in astrocyte numbers can explain this reduction and unfortunately, Figure 1 gives only relative numbers.

We have now included the raw numbers of astrocytes, ensheathing glia and cortex glia as a graph (Appendix Figure S2A). The number of glia observed in 7d old Canton S controls is 688 ± 28 anti-Repo-positive cells and in the *mir-31a* mutant is 417 ± 37 anti-Repo-positive cells. At 7d, there are 482 ± 19 alm-Gal4>UAS-Histone-RFP+ cells in the control animals (n=53) and 348 ± 11 alm-Gal4>UAS-Histone-RFP+ cells in the *mir-31a* mutants (n=35).

Thus, astrocytes account for a large number of the cells that are lost. We do not exclude that ensheathing, cortex, subperineurial and perineurial glia might be affected to some extent, but loss of astrocytes appears to account for the majority of the total glial loss observed. Note that the number of astrocytes is not significantly different at 21d post-eclosion between *mir-31a* mutant animals and control animals (Figure EV1D and Appendix Figure S2B)

The notion that mir-31a is required in the astrocyte progenitors to allow survival of the differentiated cells is not really solved. In fact, the adult depletion of mir-31a in adult progenitor cells also results in the loss of adult glial cells - but at this developmental stage all glial cells have been born already. How can the cell death be explained?

The finding that depletion of *mir-31a* leads to transient loss of adult glia shows that the population of glia is not static. The number of glia does not change much under normal conditions, so it may appear that all the glia are cells “that have been born already”.

Our findings indicate that this apparently stable situation reflects a dynamic process in which cell number is actively maintained and new cells are produced to maintain the steady state. We suggest that loss of glia in the mutant reflects a normal process of turnover (cell death and replacement). In other words the loss of cells seen in the mutant results from failure to replace glia at the normal rate. The mature glial cells are not dying due to absence of *mir-31a*: reducing the amount of *mir-31a* in glia with the sponge does not change glial numbers (Figure 2A). Rather, in the absence of *miR-31a* the replacement of glia by the progenitors is inefficient, because more of their progeny die (due to aberrant inheritance of *Rchy1* from the progenitors). Lower than normal replacement rate leads to (transient) net loss of differentiated glia.

The expression of mir-31a is not well documented and the images are not instructive. It would be indeed interesting to see the localization of astrocyte progenitors are located.

To address this, we looked at the expression of *mir-31a* in 1d old adult brains using *mir-31a* sensor flies, which carry two copies of the reverse complement of the mature *mir-31a* sequence downstream of GFP expressed under the control of the tubulin promoter. In these flies, cells that lack *mir-31a* are GFP-positive and cells that express *mir-31a* are GFP-negative. We crossed the sensor into the *Insc-Gal4>UAS-Histone-RFP* background. Cells that were GFP-negative and RFP-positive indicate the position of the progenitors that are largely astrogenic (Figure EV 1F).

We also looked at the position of the *Alrm-Gal4>UAS-Histone-RFP* cells in the *mir-31a* sensor background. We identified RFP-expressing and *mir-31a*-expressing cells (Appendix Figure S6 B, C). We note that the position of both the *Insc-Gal4>UAS-Histone-RFP*-expressing, GFP-negative and *Alrm-Gal4>UAS-Histone-RFP*, GFP-negative cells are in approximately the same location within the brain.

The generation of adult astrocytes was previously analyzed by Awasaki et al., but this paper is unfortunately not discussed.

Thanks for pointing this out. We have added a comment on Pages 4, 5 and 10.

The present study only uses Repo staining to determine cell numbers and does not address the shape of the astrocytes. In a previous report, Stork et al., (2014) had demonstrated that upon induction of cell death in astrocytes, the surviving cells expand their size. Is this observed in mir-31a mutants as well? And is this process reversed in the regeneration phase?

We appreciate the reviewer’s interest in this, but think this addresses an issue beyond the scope of this study.

Referee #2:

In "Defects in astrocyte production in mir-31a mutants unveil glial homeostasis in the adult Drosophila brain," Foo and colleagues describe a mechanism for continued glial turnover in the mature fly brain by newly identified mir-31a-positive neuroglial progenitors. The authors demonstrate that reduced mir-31a function in global mutants and by RNAi KD in progenitor cells resulted in transient loss of brain glia. While mir-31a was not required for initial production of glial cells, they found that glial cells were lost over time due to apoptosis, and that cell death occurred due to impaired post-transcriptional regulation of the E3 ubiquitin ligase Rchy1 by mir-31a. Interestingly, glial cell loss was followed by a period of biased glial cell production from mir-31a-positive cells, suggesting a homeostatic mechanism by which glial cell numbers are closely monitored and maintained within the adult brain.

Overall, the data that the authors present is clear, and the importance of the work is quite high. The

authors should be commended for the depth of their analysis. Yet there are several important changes/additions necessary before it is suitable for publication in EMBO J.

Major comments:

1. What is the "cell of origin" for the replacement glia?- glia? neurons? progenitors? other? The evidence that they come from mir-31a-gal4 positive cells is not sufficient to identify the cell type that produces these late-born glia. For example, on page 16 it says "production of new neurons and new glia from progenitor cells" - what is the evidence that these new cell come from neuroblasts (or any known progenitor)?

We think the issue here is mainly a semantic one. The clonal analysis in Figure EV3D shows that *Insc-Gal4* expressing cells make neurons and glia. Likewise, cells that express *mir-31a-Gal4* divide to make glia and (to a lesser extent) neurons (Figure 4C,D). In other words, by virtue of the fact that they divide to make new progeny in the adult, both the *Insc*-expressing and the *mir-31a*-expressing cells are functionally defined as progenitor cells.

The *Insc-Gal4*-expressing cells are mainly neurogenic, but also make some glia. The *mir-31a* cells are mainly gliogenic, but can make some neurons. In our view, there is no better way to define these cells as progenitors than to show that they divide to make glial and/or neuronal progeny in the adult. We don't know of other markers that would help to better define the identity of the adult progenitor cells that make neurons and glia.

2. In general, the authors provide many graphs to support their conclusions, but not enough representative images of these data. To feel confident in their conclusions, representative images for the following experiments should be displayed. If space limitations are a problem, some of the images could be included as supplemental figures.

We have now included representative pictures for all genotypes used in the Appendix Supplemental Figures S1, S2, S4, S5, S6, S8, S9 and S10.

a) Figure 1C. The paper highlights that these data suggest a mechanism for adult astrocyte homeostasis that hasn't been clearly demonstrated before, so it is essential to show that astrocytes are specifically affected, using an astrocyte specific marker.

Analysis of *mir-31a* mutants showed that astrocytes were affected but other glial types were not (Figure 1C). *Alrm-Gal4* was used to label astrocytes in this experiment. At 7d, there are 482 ± 19 *alrm-Gal4 > UAS-Histone-RFP+* cells in the control animals ($n=53$) and 348 ± 11 *alrm-Gal4 > UAS-Histone-RFP+* cells in the *mir-31a* mutants ($n=35$).

It is clear from the numbers that loss of astrocytes can account for the majority of the total glial loss observed, but we do not exclude that ensheathing, cortex, subperineurial and perineurial glia might be affected to some extent that is not significantly different. We have modified the text on Page 6 to emphasize this.

b) For images comparing the mutant phenotype to loss of glia due to use of mir031a sponge in insc-Gal4 (Fig. 2D) and mir-31a-Gal4 (Fig. 2F) animals, as they are used as proxies for the global mutant. Because the data is always presented as % of the control, it is not possible to determine how well the cell-specific KDs reflect the mutant phenotype.

Raw numbers are now provided as graphs in the Appendix Supplemental Figures. Specifically, the raw data graph for Fig2D is Appendix Figure 3D and representative images are in Appendix Figure S4 and S5. For Fig 2F the raw data graph is Appendix Figure 3F and the representative images are in Appendix Figure S5 and S6.

c) For experiments where the authors suppress the mutant phenotypes either by blocking apoptosis (Fig. 2D) or by providing exogenous mir-31a (Fig. 2H)

The raw data graph for Fig2D is Appendix Figure 3D and representative images are in Appendix Figure S4 and S5. For Fig 2H, the raw data graph is Appendix Figure 3H and the representative images are in Appendix Figure S5 and S6.

d) Fig. 3A, C to definitively show mutant phenotype in response to CG16947 manipulations

The raw data graph for Fig 3A is in Appendix Figure S7B and the representative images in Appendix Figure S8.

The raw data graph for Figure 3C is in Appendix Figure S7D and the representative images in Appendix Figure S8.

3. The authors should use a "progenitor specific" *gal4* line, such as *wor-gal4* in addition to *insc-gal4* (which is expressed in some glia; Omoto et al., 2015). Showing both give the same phenotype, for at least a few experiments, would help prove that the effects are neuroblast specific.

We now include data where we depleted *mir-31a* using the *mir-31a* sponge with *Worniu-Gal4*. The *Worniu-Gal4*>*UAS-31a* sponge recapitulates our *Insc-Gal4*>*UAS-31a* sponge result (Appendix Figure S6D, E, S9).

4. page 16, are the *EdU*⁺ glia endoreplicating or forming *de novo*?

Glial cells are lost in the mutant, and the number of cells subsequently recovers. The MARCM clonal analysis also shows that new cells are made by division of progenitors during the recovery process. The *EdU* incorporation presumably reflects the production of these new glia. We do not exclude that there might also be some endoreplication (for example in perineural glia).

Minor issues

- please provide more information in Table S1 (e.g. *p*-values, primers used for all genes) and in the methods regarding their qPCR conditions, standardizations and results, which should follow MIQE guidelines (PMID 19246619)

We have included information on qPCR conditions in the methods section on Page 24 in accordance with the MIQE guidelines. The sequences of the primers are in Table S1.

- authors should not use the term 'astrocytes' unless cell identity is confirmed with an astrocyte marker (e.g. *alm-gal4*)

We have changed the manuscript accordingly.

- The authors show that overexpression of CG16947 results in reduced glial cell number, but they don't directly show that this is due to apoptosis (as was shown for loss of *mir-31a*). Please stain for a cell death marker to show that overexpression of *Rachyl* mimics the *mir-31a* mutant phenotype.

We have overexpressed *UAS-CG16947* using *Repo-Gal4* and find the presence of activated caspase 9, anti-*Repo*-expressing cells in 2d old adult brains (Figure EV2 E).

- Whenever presenting data, please include +/- SD or SEM. For example, on page 5, "dropped to approximately 62% of Canton S controls" how many *n*'s and what is the range of glial numbers are not shown.

We used scatter plots so that "n" would be evident for each sample. Error bars in the graphs show SEM. Phrasing in the text sometimes used less precise language. To address the reviewer's request we have compiled a table with SEM and sample size in numerical form (Table EV2).

- Throughout the text and figure legends, please be more clear about whether cells were counted by reporter expression or by antibody stain. For example, the authors use both *repo-Gal4* and anti-*Repo*, and it's not always clear what method was used for counting

We have changed the text and figure legends to indicate 'anti-*Repo*' where we counted cells labeled by antibody. Where we counted cells visualized with a *Gal4* driving the expression of *UAS-Histone-RFP*, we have indicated as such.

- Fig. 4E, please state what animals and what time point was used for this quantification

We have added this information in the figure legend of Fig 4E, Page 37.

- Please label each figure panel with the time point analyzed

Done

- The wording in the discussion is at times too strong based on the data the authors provide. For example, on page 17, the authors intimate that the *mir-31a* progenitors are specifically astrogenic. Although the data that they provide shows that glial survival is specifically affected by loss of *mir-31a*, these data do not preclude that *mir-31a* progenitors also give rise to cortex and ensheathing glia and that they simply do not have a survival phenotype.

We counted the number of cortex glia, ensheathing glia and astrocytes and only observed a significant defect in the number of astrocytes in the central brain. This suggests that *mir-31a* progenitors affect astrocyte survival. We cannot exclude the possibility that *mir-31a*-expressing progenitors also give rise to cortex and ensheathing glia, but if so, it appears that inheritance of *rchy1* by these cells does not lead to loss of their progeny. We have added a comment to the text to note the reviewer's caveat on Page 17.

2) For example, on page 19, glial turnover has been shown in the healthy brain during learning/memory tasks in rodents (reviewed in PMID 4634839).

Thanks for pointing this out. It has been noted in the discussion on Page 20..

- the authors should carefully review and edit genetic nomenclature throughout the text and figures (e.g. use of lowercase italics whenever genes and transgenes are used for *Drosophila* versus rodent genes)

Very minor issues:

- Consistency with labeling in the figures

Done

- Make sure to define transgenes when they're first presented. For example, *repo-Gal4* is first defined on page 7, but introduced on page 6

Done

- Typo on page 11 "labeled after 1 day with anti-Repo to mark neurons (Figure 4A) or glia (Figure 4B, S2A). The authors used *Elav* to mark neurons, *Repo* to mark glia
Fixed. Thanks for catching this!

- Fig. 5J+K are cutting off parts of the labels for Fig. 5H + I

Fixed. Thanks for catching this!

Referee #3:

General Summary:

This paper by Foo and colleagues describes a novel progenitor pool in the young adult Drosophila brain that is identifiable by microRNA mir-31a expression. These progenitors give rise largely to glial cells, likely astrocytes. In the absence of mir-31a, additional adult progenitors restore glial cells to normal numbers over the course of weeks. This provides the first description of a homeostatic mechanism in the adult Drosophila brain to maintain sufficient number of glia. These results are novel and very exciting and will provide important new insight into microRNA and ubiquitin ligase-mediated regulation of glial cell numbers in adults, which will be of broad interest to many in the field of glial cell biology. Overall, the manuscript is well written and the conclusions largely supported. Some thoughts and comments to improve the paper are provided below:

Major points:

1. Figure 1A, the loss of Repo-positive cells is apparent. However, because astrocyte cell bodies are specifically located at the edge of neuropil regions, it's a little surprising that the Repo reduction appears consistent (broad) throughout the entire brain. Perhaps due to the particular image that was selected? It would be nice if confocal images were provided for some additional key experiments throughout the paper. For example, images corresponding to the Fig 2E-H. This would provide another opportunity to potentially observe the spatial distribution of changes in Repo expression throughout the central brain.

We have included representative pictures for all genotypes used in the Appendix Supplemental Figures S1, S2, S4, S5, S6, S8, S9 and S10.

2. As written, the manuscript seems to make an abrupt transition from using *alm-Gal4* to *Insc-Gal4* to visualize/manipulate glia. Is there experimental evidence showing that *alm*⁺ positive cells do not give rise the same progenitors that *Insc*⁺ (proliferative) cells do? *Insc* may not be specific to astrocytes.

We used *alm-G4* to label astrocytes in Fig 1C. We used *alm-G4* as a marker to compare astrocyte number in mutant and wild-type flies. Subsequent experiments used *Insc-Gal4* or *mir-31a-G4* to direct marker or sponge expression. The MARCM clonal experiments shows that both *Insc-Gal4* and *mir-31a-G4* cells divide to give progeny in the adult – identifying them as functional progenitor cells in the adult.

We did not suggest that *alm-G4* cells are progenitor cells, or that they give rise to them.

Note also, that we did not suggest that *InscG4* was specific to astrocytes. Quite the opposite. We provide MARCM clonal data showing that *InscG4* cells serve as progenitors for both neurons and glia.

It would nice if a subset of key expts were repeated co-labeling with the GABA transporter Gat - or the excitatory amino acid transporter EAAT1 (aka GLAST) - to show that the newly produced cells that contribute to recovery of the glial population in mir31a mutants are indeed astrocytes.

The data provided in expanded view figure E1 addressed this point. We used *Alrm-Gal4* to drive *UAS-Histone-RFP* to label astrocytes. The number of astrocytes was significantly reduced in the mutant at day 7 (compared to control) but recovered by day 21. This provides evidence that loss and replacement of astrocytes contributes to the mutant phenotype.

The antibodies recommended by the reviewer would not be suitable for cell counting, because they label the whole cells, not just the cell body. Using whole cell markers (such as *UAS-CD8-GFP*) makes it very difficult to get accurate cell counts, compared to use of a nuclear marker, like *UAS-histone-RFP*. In addition the quality of the antibodies available to us too poor to given reliably quantifiable data. We cannot do the specific experiment, but we suggest that the point is adequately addressed by the *alm-G4* marker experiment.

3. *Repo-driven expression of the mir31a sponge did not alter the number of glial cells. Repo appears to co-express with mir31a drivers quite early (during the first week). How do the authors reconcile the fact that there was no phenotype with RepoGal4 -> mir31a sponge expression?*

As the reviewer points out, removing *mir-31a* expression with the sponge in mature glia has no effect (Figure 2A). We think that the co-expression of Repo with RFP may reflect perdurance of Gal4 and Histone-RFP proteins expressed in the progenitors (*mir-31a-Gal4* drives *UAS-Histone RFP*). Using a *mir-31a sensor* transgene (GFP expression indicated absence of the microRNA), we did not see evidence of the miRNA in mature glia (Expanded view Figure EV1E).

4 . Using heterozygote H99 mutant animals may influence apoptosis during development. In Figure 2D, the authors show that the reduction of Repo-positive cells in 31a sponge flies is rescued in H99 heterozygotes. Might the H99 hets already have higher levels of glia (and perhaps other cells?) Including an H99 heterozygotes is an essential control in this experiment. Also, confirming that apoptosis is altered with an activated caspase-3 stain would be nice addition to round out this part of the model.

We have included the control genotype of *Insc-Gal4>GFP, Df(3L)H99* and see no appreciable change in glia number in these flies (Figure 2D).

We observe apoptotic glial cells (activated caspase 9-expressing, anti-Repo-expressing cells) in the brains of 4d post-eclosion *mir-31a* mutants (Figure 1E).

5 . The model of astrocyte homeostasis is intriguing one, and this paper presents the first description of this mechanism in adult *Drosophila*. The authors state that "turnover of astrocytes has not yet been observed in healthy brains of adult mammals", but I don't believe this is true. For example, Sohn et al., (*J Neuroscience* 35(9): 3756-63, 2015) describe astrocyte production balanced by apoptosis in adult mice.

Thanks for pointing this out. We have modified the text on Page 20 accordingly.

Minor points:

1 . State N values for experiments.

We used scatter plots for the quantitative data so that "N" would be evident for each sample. Error bars in the graphs show SEM. Table S2 contains the N values for all experiments in Table S2.

2 . Missing word: Depleting *mir-31a* in mature glia with *Repo-Gal4* (reversed polarity, Figure 2A) or in post-mitotic neurons WITH *synaptobrevin-Gal4* (*Syb-Gal4*, Figure 2B) did...

Thanks for pointing this out. We have modified the text on Page 7 accordingly.

3 . Change sentence: "Adult flies were heat-shocked the day of eclosion to induce clones and the brains were labeled after 1 day with anti-Elav or anti-Repo to mark neurons (Figure 4A) or glia (Figure 4B, S2A)."

Thanks for pointing this out. We have modified the text accordingly on Page 12.

4 . Fig 4C and D are incorrectly labeled "*31aGal4,G80ts>G-Trace*". There is no G-Trace in this image I think.

Done

5 . Fig 5E - are mutant and control different magnifications?

Apologies, this has been fixed.

6 . Fig 5H - What time point was this and where were the EdU cells counted? The whole brain?

These were done on 6d post-eclosion animals and the EdU cells in the central brain were counted. Added to the figure legend accordingly on Page 39.

7 . I'm assuming the experiment shown in Supplemental Figure 1E was done in a *mir-31a* mutant background? It doesn't explicitly say that in the Results or Figure/Figure legend.

This was done in wild-type flies. We wanted to verify that the Repo marker was stable in glial cells with age, to exclude a concern, raised by a reviewer of an earlier version of the manuscript, that loss of the marker could be mistaken for loss of the cells. This point could only be addressed in the wild-type background, since cells are transiently lost in the mutant.

8. Most of the images in Fig 3G,H are very difficult to see. All channels can be converted and grayscale since there are no merged images.

Done

2nd Editorial Decision

07 February 2017

Thanks for submitting your revised manuscript to The EMBO Journal. Your manuscript has now been re-reviewed by the three referees.

As you can see below, the referees appreciate the introduced revisions and support publication here. They have a number of relative minor points that I would you to sort in a final revision.

REFEREE REPORTS

Referee #1:

The authors have addressed most of the comments but neglected one, which in my view is a must before publication.

The mir-31 mutant affects glial cell number in 7 day old fly brains. The number of glia is hidden in the text but the figure still shows relative numbers. The graphs should show actual numbers. In the rebuttal letter it is stated that Canton S flies have 688 glial cells, of which 482 are alm positive. In mir-31 mutants, only 417 glial cells are found of which 348 are alm positive. In the main text it is stated that 62% of the 688 wild type glial cells are found in the mutant. This accounts for 427 cells. What is correct?

In any case, mir-31 mutants have 271 (261) fewer glial cells compared to Canton-S.

The number of alm positive cells drops by 134 glial cells. This is only about 50% of the total glial cell loss (271 or 261). A statement that loss of astrocytes accounts for the majority of the total glial loss is thus not correct and should be removed.

Since no changes in the number of ensheathing glia and cortex glia were noted, it is likely that the number of the perineurial glial cells is affected. Since these cells are generated by different means compared to astrocytes, this should be stated and discussed.

Referee #2:

The revised manuscript addresses all my concerns and is suitable for publication.

Referee #3:

The authors appear to have addressed the concerns of this reviewer in good faith, and the manuscript is suitable for publication in EMBO.

Two minor points:

1. It is unclear what the "heterozygote" versus "mutant" animals reflect in Figure 4C and D?
2. Arrowhead is missing from Figure EV4 (repo EDU merge panel).

2nd Revision - authors' response

09 February 2017

Referee 1:

The authors have addressed most of the comments but neglected one, which in my view is a must before publication.

The mir-31 mutant affects glial cell number in 7 day old fly brains. The number of glia is hidden in the text but the figure still shows relative numbers. The graphs should show actual numbers.

We feel that the differences between control and mutant conditions are more easily grasped by showing mutant relative to the control in the main figures. The “actual” numbers are given in the text and in Appendix Figure S1A, 2A. Nothing is “hidden”.

In the rebuttal letter it is stated that Canton S flies have 688 glial cells, of which 482 are alrm positive. In mir-31 mutants, only 417 glial cells are found of which 348 are alrm positive. In the main text it is stated that 62% of the 688 wild type glial cells are found in the mutant. This accounts for 427 cells. What is correct?

417 is the correct figure. The 62% number appears to be a typo. It should have said 61% (60.6%). Given the variance, we have revised this to ~60% in the text.

In any case, mir-31 mutants have 271 (261) fewer glial cells compared to Canton-S. The number of alrm positive cells drops by 134 glial cells. This is only about 50% of the total glial cell loss (271 or 261). A statement that loss of astrocytes accounts for the majority of the total glial loss is thus not correct and should be removed. Since no changes in the number of ensheathing glia and cortex glia were noted, it is likely that the number of the perineurial glial cells is affected. Since these cells are generated by different means compared to astrocytes, this should be stated and discussed.

We thank the reviewer for pointing this out. The emphasis on astrocytes was perhaps unwarranted. The important issue is loss and replacement of glia. The text has been modified to replace references to “astrocytes” with references to glial cells, except where the experiments addressing glial subtypes are presented. In this section we now summarize the results as follows: “Loss of alrm-Gal4 expressing astrocytes accounts for ~half of the missing glia. The identity of the other missing glia has not been determined.”

Referee #3:

The authors appear to have addressed the concerns of this reviewer in good faith, and the manuscript is suitable for publication in EMBO.

Two minor points:

1. It is unclear what the "heterozygote" versus "mutant" animals reflect in Figure 4C and D? "heterozygote" means the miR-31a-Gal4 mutant allele/+. "mutant" means the miR-31a-Gal4 allele/miR-31a KO allele. The figure legend has been changed to indicate this.

2. Arrowhead is missing from Figure EV4 (repo EDU merge panel). Thanks for spotting this. It has been fixed.

YOU MUST COMPLETE ALL CELLS WITH A PINK BACKGROUND ↓
PLEASE NOTE THAT THIS CHECKLIST WILL BE PUBLISHED ALONGSIDE YOUR PAPER

Corresponding Author Name: Stephen M. Cohen
EMBO Journal
Manuscript Number: EMBOJ-2016-95861